# Ancestry-specific polygenic risk scores are risk enhancers for clinical cardiovascular disease assessments

George B. Busby [1] ✉, Scott Kulm[1], Alessandro Bolli[1], Jen Kintzle[1], Paolo Di Domenico[1] & Giordano Bottà[1] ✉

Clinical implementation of new prediction models requires evaluation of their utility in a broad range of intended use populations. Here we develop and validate ancestry-specific Polygenic Risk Scores (PRSs) for Coronary Artery Disease (CAD) using 29,389 individuals from diverse cohorts and genetic ancestry groups. The CAD PRSs outperform published scores with an average Odds Ratio per Standard Deviation of 1.57 (SD = 0.14) and identify between 12% and 24% of individuals with high genetic risk. Using this risk factor to reclassify borderline or intermediate 10 year Atherosclerotic Cardiovascular Disease (ASCVD) risk improves assessments for both CAD (Net Reclassification Improvement (NRI) = 13.14% (95% CI 9.23–17.06%)) and ASCVD (NRI = 10.70 (95% CI 7.35-14.05)) in an independent cohort of 9,691 individuals. Our analyses demonstrate that using PRSs as Risk Enhancers improves ASCVD risk assessments outlining an approach for guiding ASCVD prevention with genetic information.

Atherosclerotic Cardiovascular Diseases (ASCVD) contribute an increasing burden on healthcare systems and are responsible for 1 in 3 deaths worldwide[1]. Whether an individual will suffer from ASCVD depends on the complex interplay between environmental and life-style exposures such as poor diet and lack of physical activity, physiological components such as high LDL cholesterol and high systolic blood pressure[2] and their genes. Given the high heritablity of ASCVD (40–60%)[3] it is unsurprising that many studies have successfully uncovered genetic contributions from both rare pathogenic mutations in genes such as *APOB*, *LDLR* and *PCSK9*, that can cause hypercholesterolemia, and more common genetic variants spread throughout the genome[4–8]. Utilizing common genetic variation, Polygenic Risk Scores (PRSs), have shown promise as tools to quantify genetic risk and identify significant proportions of the population at high genetic risk of cardiovascular disease[9–12].

However, questions remain about the utility of PRSs in clinical care. These include doubts and misconceptions about what PRSs are and how they should be used[13], as well as important considerations about the lack of transferability across populations with divergent ethnicities and genetic ancestries from those used in their development[14]. Historical biases in the collection and analyses of genomic data have led to an over-representation of individuals of European genetic ancestry in clinical datasets[15], which in turn has led to a lack of sufficiently diverse data with which to build PRSs. While the attenuation of PRSs in different populations is expected based on known differences in allele frequencies and linkage disequilibrium patterns across global populations[16], for PRSs to be utilized clinically they need to deliver meaningful risk stratification regardless of the range of genetic ancestries present in the populations in which they are intended to be used.

Here we describe the development and validation of multi-ancestry PRSs for CAD using an approach that utilizes multiple ancestry-specific GWASs[17] and diverse PRS discovery, validation, and testing datasets. The aim of this work is to provide evidence of the performance of CAD PRSs across diverse ancestries and to demonstrate how they can be used to enhance clinical ASCVD prevention tools. Throughout, we use the term *genetic ancestry* and *ancestry* interchangeably to refer to groupings of individuals who are similar to each other genetically and label these groups using current-day geographical names that relate to the continents on which most

[1]Allelica Inc, 447 Broadway, New York, NY 10013, USA. ✉e-mail: george@allelica.com; giordano@allelica.com

individuals belonging to that group currently reside. These labels are intended to assign individuals to groups based on their genetics alone and are not social or ethnicity identifiers. We acknowledge that these labels are imperfect, both because modern human populations rarely contain ancestry from a single region[18], but also because they enforce the artificial discretisation of continuous human genetic diversity[19]. Moreover, when grouping individuals into continental-level ancestries, as we do here, a large amount of within-group diversity is concealed. Nevertheless, because genetic variation is shared amongst individuals with similar genetic ancestry, these approximations help to assess genetic effects in groups of similar individuals, and are a necessary tradeoff between optimizing PRSs for a diverse range of populations and having sufficient data from such groups to validate and test resulting scores.

## Results

We developed 152 PRSs for CAD by applying 38 different combinations of seven separate ancestry-specific GWASs, including a range of allele frequency filters and finemapped variations (Supplementary Table 1, Supplementary Figs. 1 and 2 and "Methods"), to the PRS-CSx tool[17] implemented within Allelica's *DISCOVER* software. Fine-mapping is particularly appropriate for this study because it increases the portability of GWAS signals across populations by identifying putatively causal as opposed to linked variants, and has been shown to increase the transferability of PRSs across ancestries[20]. We used three publicly available prospective cohorts comprising clinical ASCVD related covariates and matched genetic data to validate and test these PRSs: the MultiEthnic Study of Atherosclerosis (MESA);[21] Atherosclerosis Risk in the Community (ARIC);[22] and UK Biobank (UKB)[23]. These data were harmonized into a combined dataset of 30,809 cases and 466,860 controls of which 27,158 cases and 440,404 controls were employed in genome wide association studies and 2931 cases and 26,456 controls were employed in PRS development (Supplementary Table 2). We inferred continent-level genetic ancestry proportions for each individual and assigned them to genetic ancestry groups on the basis of this inference ("Methods"). We applied the 152 PRSs to five genetic ancestry-specific PRS validation datasets, both individually and in combination as metaPRSs, so that the best performing score in each genetic ancestry group could be identified (Supplementary Table 3). To the best of our knowledge, the output of PRS-CSx has never been previously used to create a metaPRS in this way. These best scores were then evaluated in independent ancestry-specific testing datasets to define a final assessment of their predictive performance (Fig. 1). When possible, we focused on fixed effects meta-analyses of the results of two independent cohorts of a single ancestry group. The predictive performance measured by the score's Odds Ratio per Standard Deviation (ORxSD) ranged from 1.47 (95% CI 1.08–2.01) in the American genetic ancestry group to 1.81 (95% CI 1.31–2.50) in the South Asian genetic ancestry group (Fig. 1a).

To contextualize the performance of the novel multi-ancestry scores we benchmarked them against three previously published scores[9,10,12]. Our multi-ancestry PRSs had a greater ORxSD than previously published panels across all ancestry groups (Fig. 2 and Supplementary Tables 4 and 5). As an illustrative example, models fit to predict incident and prevalent cases of CAD in individuals of African ancestry in the UK Biobank resulted in an ORxSD of 1.37 (95% CI 0.935–2.01) and for individuals in MESA resulted in an ORxSD of 1.79 (95% CI 1.14–2.81). The ORxSD from a meta-analysis of these two results was 1.53 (95% CI 1.15–2.05), greater than ORxSDs of 1.04 (95% CI 0.79–1.39) from the GPS_CAD PRS, 1.24 (95% CI 0.92–1.67) from the Allelica_CAD_EUR_2020 PRS and 1.30 (95% CI 0.96–1.76) from the multiGRS_CAD PRS (Supplementary Table 6). These differences suggest that not utilizing the novel PRSs for future use would likely lead to relatively inferior predictions.

As a major aim of this study is to build accurate models across ancestries it is necessary to test calibration as well as model performance. Our multi-ancestry PRSs were similarly better calibrated (lower Brier score) than competing scores for all single ancestry groups (Supplementary Table 6) as well as in a group of individuals who were of admixed ancestry and could not be assigned a single ancestry label. In this group, we used the ancestry weighted sum of the single ancestry model predictions to compute a Brier score of 0.06085 for all of the Allelica_CAD_vJ PRSs ("Methods"). The Brier scores computed in an equivalent process for the GPS_CAD score was 0.06089, Allelica_CA-D_EUR_2020 PRS was 0.06095 and multiGRS_CAD PRS was 0.06107 (Supplementary Table 6). We hypothesize that this increase in calibration performance compared to all alternative PRSs was due to the additional information gained from the ancestry-specific effect sizes present in the multiple GWASs and the use of finemapping to focus effects towards causal variants. Additional statistics, such as Nagelkerke $R^2$ and the Area Under the receiver operator Curve, and models containing additional sets of covariates, such as age and/or sex alone, further confirmed that our novel multi-ancestry score performed better than the alternative scores (Fig. 2, Supplementary Figs. 3 and 4, and Supplementary Tables 4 and 8).

The Pooled Cohort Equations (PCE)[24] are a validated clinical tool that take multiple risk factors associated with ASCVD to calculate an individual's 10-year risk of disease[25]. The PCE and other related ASCVD risk prediction algorithms do not currently take genetics into account despite growing evidence demonstrating both that individuals at heightened risk of disease can be identified by PRSs, and that these individuals are largely invisible to clinical risk assessments[9–11,26,27]. Current American Heart Association and American Academy of Cardiology blood lipid management guidelines advocate that for individuals with borderline (5–7.5%) or intermediate (7.5–20%) 10-year risk (BIR), additional Risk Enhancers can be used to further refine risk mitigation discussions between physician and patient, and PRSs have recently been proposed to aid clinical management of ASCVD in such discussions[25,28] while reducing healthcare costs from a payer's perspective[29]. We therefore sought to assess the potential for the multi-ancestry PRSs to be used as a discriminatory risk enhancing factor for CAD by using a twofold increased risk threshold to identify individuals at high risk of CAD. Similar to the ORxSD, this threshold carries a range of error that is exacerbated by the size of the sample analyzed. Nevertheless its point estimate is equivalent to that of well established risk factors such as family history and some Mendelian-inherited genetic variants[30] as well other risk enhancing factors such as diabetes and ethnicity which are currently used to upclassify BIR PCE individuals[11,24,31].

In a PCE testing dataset comprising 9691 individuals of diverse ancestry, we identified ancestry-specific twofold risk thresholds as the percentile of the PRS distribution at which the risk in the upper tail is twice the risk of the lower tail ("Methods"). With meta-analyses where possible, we identified the top 12–24% of individuals of each ancestry group with twofold increased risk compared to the remainder (Fig. 1c, d and Fig. 2c). We tested the accuracy of these thresholds by applying them to a unused subset of 6480 admixed individuals from the testing dataset by constructing unique individual-level thresholds as an average of the ancestry-specific percentiles weighted by each admixed individual's ancestry proportions ("Methods"). This approach allowed us to classify admixed individuals whose PRS scores placed them at twofold increased genetic risk. We then tested whether this binary risk factor recovered twofold increased risk for CAD for those carrying it with logistic regression, and confirmed that the thresholds were able to accurately assign individuals at twofold risk in this group (ORxSD of 1.96; 95% CI 1.42–2.70). In addition to accurately assigning risk in the independent admixed group, the percentile thresholds also identified a greater share of individuals than any of the competing PRSs. For example, the average difference between the twofold threshold found

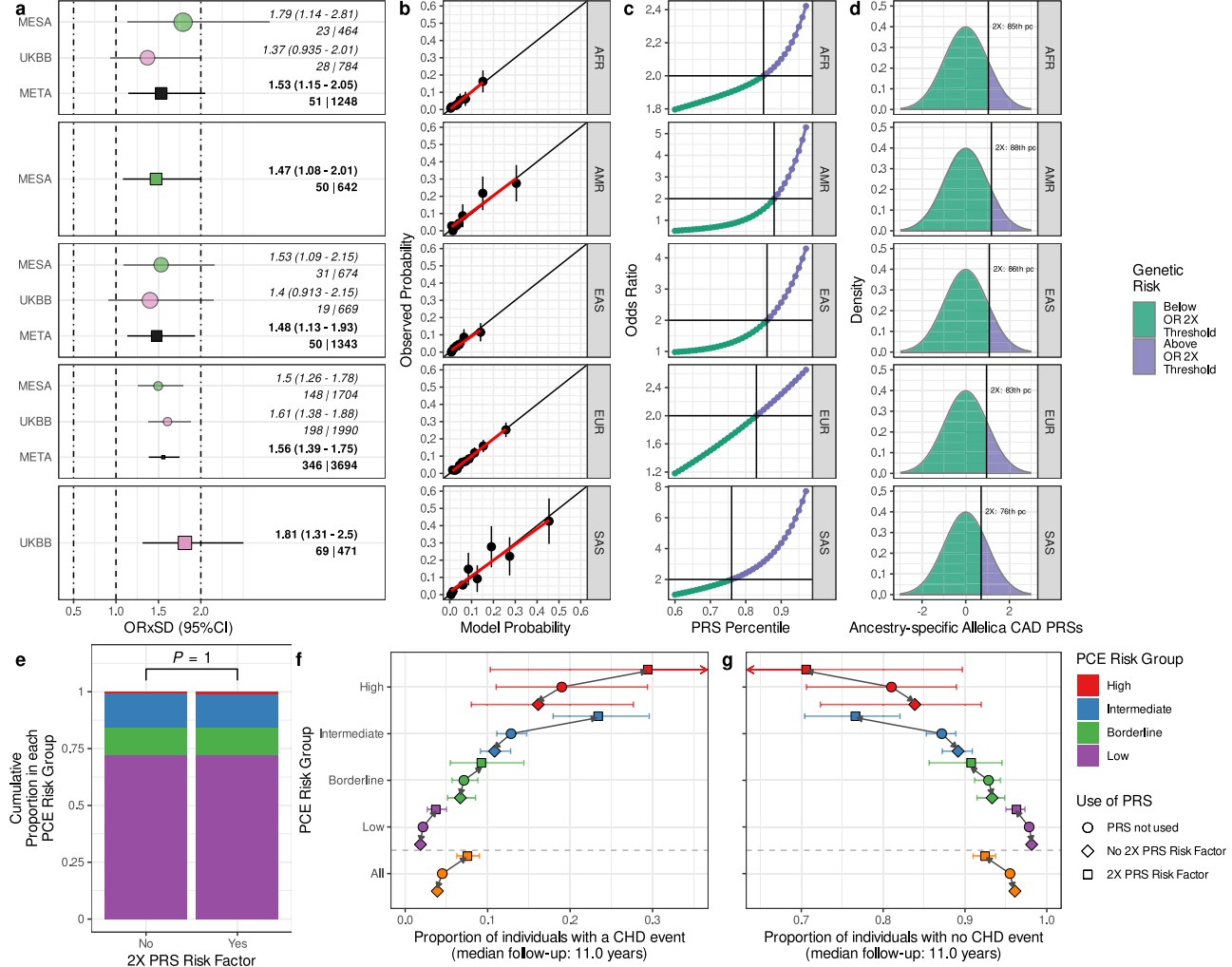

**Fig. 1 | Multi-ancestry PRSs for CAD predict risk in diverse ancestries and add to clinical risk assessments.** For five different genetic ancestry groups we show **a** the point estimate and 95% CI of the Odds Ratio per Standard Deviation (ORxSD) of the ancestry-specific polygenic risk scores in the UK Biobank and MESA Testing populations. META describes the results of a meta analysis of the cohort-specific ORxSDs. Squares denote the reported ORxSD for each ancestry and point size is proportional to the standard error of the ORxSD estimate. Values on the right of the panel are the reported ORxSD with 95% CI together with the numbers of cases and controls in the Testing populations. **b** The calibration curves of the PRS models that compare each model's response type predictions to the actual prevalence of disease. The points show the mean value of each quantity across deciles of the predictions, vertical error bars show the 95% CI around the mean probability in each decile, the red line follows the trend of these decile values and the black line shows perfect calibration; **c** the Odds Ratio based on the upper tail compared to the remainder of the distribution and the percentile threshold at which 2× increased risk is met; **d** the proportion of the PRS distribution at at least twofold increased risk

compared with the remainder; **e** comparison showing no significant difference (calculated via one-sided Fisher's Exact Test) between the proportions of individuals in each of the PCE risk groups in those with and without PRS as a risk enhancing factor; **f** the proportion of individuals in each of the PCE strata who had a CHD event during the follow-up time of the prospective cohort that they were sampled from. We show the proportions for individuals classified into each PCE group and the total dataset (All) split into individuals with and without PRS as a Risk Enhancing Factor. Horizontal error bars denote the 95% CI around the proportion of individuals in each risk group; **f** is the same as **g** except showing the proportion of individuals in each of the PCE strata who did not have a CHD event during the follow-up time of the prospective cohort that they were sampled from. Horizontal error bars denote the 95% CI around the proportion of individuals in each risk group. Total sample sizes for each group are (cases/controls): AFR / African (51/1248); AMR / American (50/642); EAS / East Asian (50/1343); EUR / European: (346/3694); SAS / South Asian (69/471).

with all of the Allelica_CAD_vJ scores and the lowest percentile threshold of each of the three competing scores (Supplementary Tables 3 and 8) averaged 7.0% (SD 3.74%). This difference would imply that for every 1000 individuals screened the Allelica scores would identify 70 more individuals at high risk who would have otherwise been missed.

We used the PCE to classify individuals with BIR of ASCVD and performed two analyses using the CAD PRS as a risk factor to reclassify individuals using separate disease endpoints of Coronary Heart Disease (CHD; $N = 9691$) and ASCVD ($N = 9569$). There was no significant difference in the proportion of individuals in each PCE risk strata with and without the twofold risk factor (Fig. 1e) indicating that the PCE alone does not capture the increased risk

attributable to the PRS risk factor. Across BIR individuals, 16% had the twofold PRS risk factor. Among BIR individuals with the twofold risk factor, there were 64% more CHD events and 47% more ASCVD events than in those BIR individuals without the PRS risk factor (Supplementary Tables 10 and 11). Upclassifying BIR individuals on the basis of the PRS risk factor resulted in a Net Reclassification Improvement (NRI) of 13.14% (95% CI 9.23–17.06%) when applying the CAD PRSs as a risk enhancing factor with CHD as an endpoint (Supplementary Table 9) and 10.70% (7.35–14.05%) when using ASCVD as the primary endpoint of the analysis (Supplementary Table 10). We confirmed that the twofold increased risk effect of the CAD PRSs was present in the PCE testing dataset (OR = 2.03 95% CI

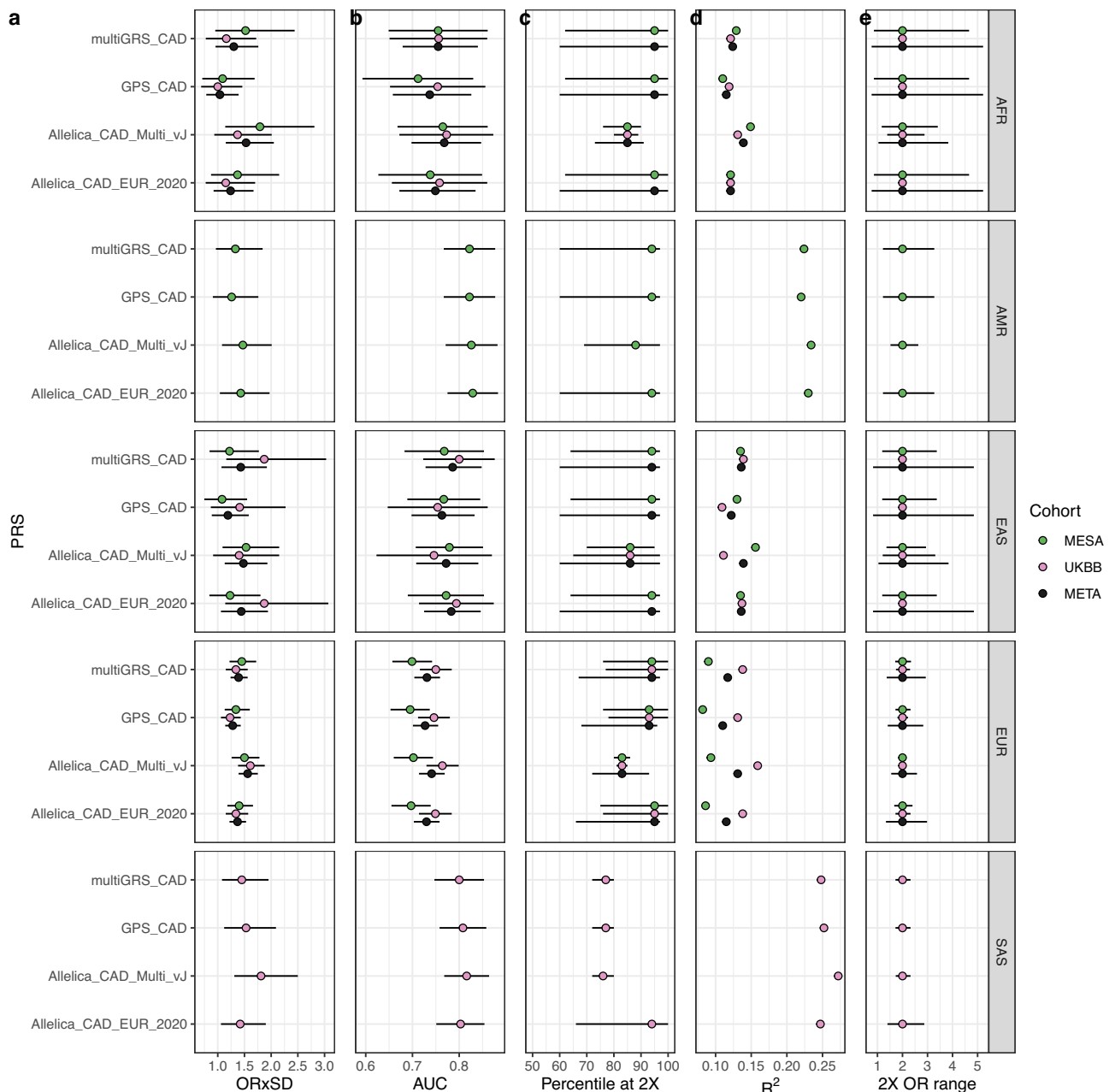

**Fig. 2 | Evaluation of the performance of the Allelica CAD multi-ancestry PRSs against three published PRSs.** For each PRS/genetic ancestry combination we show benchmarking results for the Allelica_CAD_vJ PRSs and three published PRSs applied to exactly the same testing populations. Total sample sizes of the testing populations are (cases/controls): AFR / African (51/1248); AMR / American (50/642); EAS / East Asian (50/1343); EUR / European (346/3694); SAS / South Asian (69/471) and a breakdown per sub-cohort shown in Supplementary Table 15. We assessed the statistics on the sub-cohorts and, where possible, a meta analysis of the sub-cohort values. The statistics shown, with horizontal error bars detailing the 95% CI of these values, are **a** the point estimate of the Odds Ratio per Standard Deviation of the PRSs; **b** the Area Under the Receiver Operator Curve with 95% CI estimated; **c** the threshold percentile of the PRS distribution where 2× risk is achieved; **d** Nagelkerke $R^2$; and **e** the range of the odds at the 2× threshold.

1.49–2.76) even after controlling for other PCE covariates with logistic regression (Supplementary Tables 11 and 12). Stratifying CHD cases within each PCE risk strata by the PRS risk factor confirmed that individuals with the PRS risk factor had around twice as many events than those without (Fig. 1f, g). Finally, we assessed the improvement in classification of integrating the CAD PRSs directly into the PCE (Supplementary Table 13) and observed a lower but significant categorical NRI for both CAD (NRI = 4.31%; 95% CI 1.85–6.78) and ASCVD (NRI = 2.90%; 95% CI 0.61–5.18).

## Discussion

Here we have demonstrated that validated and calibrated ancestry-specific PRSs for CAD can be used as a risk factor to better classify an individiual's overall risk of ASCVD. Nevertheless, there are several limitations to this study. First, following best practice[32] we developed and validated PRSs using independent PRS training, validation and testing cohorts which led to relatively small sample sizes, particularly for the East Asian and American testing groups. Second, these sample sizes necessitated the approach of grouping individuals into continental-level genetic ancestry groups. This is sub-optimal because

it masks genetic diversity and structure within continents which may contribute to differential disease susceptibility within such groups[33,34]. We also acknowledge that the individuals present in the non-European genetic ancestry groups were sampled from individuals currently residing in Europe and the USA and so the applicability of these multi-ancestry PRSs to individuals living in Asia and Africa remains to be tested[35]. Third, while differences in the predictive performance across genetic ancestries may be the result of linkage disequilibrium patterns or may reflect true differences in genetic architecture they may also be affected by multiple other factors, such as age and sex distribution, disease definitions, sample ascertainment, as well as variation in environmental risk factors[36]. Future work that jointly considers the environmental and genetic effects of ancestry differences on risk predictions could also provide a potential avenue for further developing the approaches outlined here. Fourth, we have concentrated on an analysis of using PRS as a Risk Enhancing Factor for the PCE. While the applicability of this approach to other risk algorithms requires further investigation, there is no reason to believe that these results are not generalizable to any ASCVD risk algorithm that uses well established clinical risk factors. Alternative approaches, such as integrating CAD and other PRSs into these risk models are also possible[37], but their development is contingent on suitably powered testing datasets. Moreover, if we view PRS as neither a deterministic nor diagnostic test for ASCVD but as a biomarker equivalent to others currently in use, then using such information to guide risk discussions has a precedent in national guidelines with Coronary Artery Calcium scoring, which is currently available to refine risk estimates for individuals at borderline/intermediate 10-year ASCVD risk[25]. It also captures the essence of how PRS should be used: as one tool among several that can be combined to understand an individual's overall risk of ASCVD.

Recent publications have highlighted the lack of diversity in genetic studies and the poor transferability of polygenic scores across populations[14,35,38]. These studies rightly advocate for the increased collection and analysis of diverse genomes to aid the clinical application of genomics in healthcare, a mission that we wholeheartedly support. However, unlike the PRSs described here, not all clinical tools currently utilized have been validated across all populations, and this has not precluded their broad use. As an example, most 10-year ASCVD risk models, including the PCE, have not yet been clinically validated in individuals of South Asian descent (and have been shown to underpredict risk in such groups[39]). While the ultimate aim should always be to develop clinical tools that work equally well across groups, pragmatism should lead us to use those tools that can be considered good enough to significantly enhance preventative efforts to ensure that their utility can be realized. In the context of this study, this means utilizing independently validated ancestry-specific PRSs that are able to identify individuals with higher genetic risk. These PRSs provide actionable information that can not be gleaned from other tests available in the current standard of care, even if there may be marginal differences in model performance across groups. We show here that multi-ancestry CAD PRSs can play a role in risk assessments by reclassifying significant proportions of individuals at increased risk of disease because of their genetics, but who are invisible and therefore missed by current ASCVD risk assessments and may nevertheless go on to have disease. These PRSs are validated and calibrated on US populations and given their demonstrated cost-effectiveness[29,40] it is increasingly clear that they satisfy the criteria to be considered a Risk Enhancing Factor in the management of ASCVD risk.

## Methods

### Association data

**GWAS.** We identified three large GWAS from the literature that did not include any individuals in the downstream Validation and Testing datasets that we used to develop multi-ancestry PRS. These were the CARDIoGRAMplusC4D GWAS[41], the Japanese 161k and the Japanese 52k studies[42,43]. The summary statistics for all of these studies were downloaded from the GWAS Catalog (Supplementary Table 2).

In the absence of publicly available South Asian and African specific GWAS summary statistics, we performed GWAS on separate subsets of individuals from both ancestries in the UK Biobank using *fastGWA*[44]. The South Asian GWAS comprised 932 cases and 4043 controls and the African GWAS comprised 91 cases and 3230 controls. *fastGWA* was used due to the computational efficiency of the generalized linear mixed model (GLMM)-based method that it implements, which has improved statistical properties when applied to binary traits. In addition we also performed a secondary GWAS on a subset of individuals from the UK Biobank with European ancestry (26,135 cases and 433,131 controls) to generate an additional set of European-based variant associations.

**Finemapping.** GWAS provides an estimate of the size and significance of the associations of alleles and disease at millions of sites across the genome. Even so, a variant that surpasses a significance threshold may not directly cause the disease of interest but rather only tag the causal variant by being in linkage disequilibrium (LD) with it. To identify putatively causal variants, we applied *POLYFUN*[20,45], a method that uses ancestry-specific LD scores and additional data on the putative function of variants across the genome to generate a set of finemapped summary statistics. We applied *POLYFUN* to the CARDIoGRAMplusC4D summary statistics together with functional information from Gazal et al.[46] and LD maps provided with the software, with a causal parameter of 1 and maintaining all other parameters at their default value. We only applied finemapping to the CARDIoGRAMplusC4D summary statistics because it allowed us to introduce functional information into our investigation with the best chance of utility, owing to the CARDIoGRAMplusC4D's great statistical power, without creating too many, unwieldy modified summary statistics.

The combination of three publicly available GWAS (CARDIoGRAMplusC4D, Japanese 161k, Japanese 52k), three self-computed GWAS (UK Biobank African, UK Biobank South Asian, UK Biobank European) and one finemapped GWAS (CARDIoGRAMplusC4D, finemapped) resulted in a final list of 7 different sets of summary statistics for downstream polygenic risk score panel creation (Supplementary Table 1).

**Variant filtering.** A series of quality control steps performed by Allelica's *DISCOVER* software were applied to each of the GWAS summary statistics prior to their use in generating PRS. Specifically, we removed all non-biallelic variants, removed variants with a minor allele frequency (MAF) less than one percent (according to the ancestry-specific 1000 Genomes reported allele frequencies[47]), and removed variants with duplicated IDs. For variants that underwent finemapping we also removed all ambiguous variants and those with non-canonical allele identifiers. For the European GWAS, we generated additional filtered datasets that only contained the 2 million variants with the lowest *P*-values, as well as a filtered set that only contained variants with MAF > 0.00001 (Supplementary Table 1). We implemented these filters to remove variants whose imbalanced allele counts lead to biased statistics.

### Developing PRS with *DISCOVER*

Polygenic risk scores are computed as a sum of single variant risk effects. The variance explained by the resultant score can theoretically reach the SNP-level of heritability. However, in practice factors, such as limited sample size, prevent the score from reaching even close to the heritability limit[14,48,49]. Determining the variants and the corresponding effects that create the most accurate polygenic risk score is an area of active research. We used *PRS-CSx*[17] a Bayesian approach that shrinks the GWAS computed effects of variants to disentangle the impact of

linkage disequilibrium while considering multiple ancestral groups, to build ancestry-specific PRSs for this study.

We implemented *PRS-CSx* within Allelica's *DISCOVER* PRS development software. The purpose of the *DISCOVER* software is to take in summary statistics and output a panel of variants, alongside effect sizes, that can be used to create a polygenic risk score of superior accuracy. This design matches the design of other well known genetics software, like *LDPred* and *PRS-CS*. In this specific investigation, within *DISCOVER* we combined GWAS summary statistics that were generated from different ancestral groups so that the modified variant effects in the output would benefit from a diversity of data. Specifically, *PRS-CSx* generated a separate PRS panel for each set of GWAS summary statistics. These PRS panels were combined into a single panel for downstream validation and testing, by combining the constituent variant effect sizes across panels using an inverse-variance-weighted meta-analysis of the panel specific posterior effect sizes. For example, we may input the EUR-1 (from CARDIoGRAMplusC4D), EUR-3 (from UK Biobank European) and AFR-1 (from UK Biobank African) GWAS summary statistics into the *DISCOVER* software, which are jointly analyzed and output as three separate PRS panels. These are then ultimately joined into a single panel of modified variants effects. We used 38 different combinations of multiple input GWAS in this way and named the resulting panel by the GWAS datasets used to generate it, so in this example the panel was named EUR-1;EUR-3;AFR-1 (Supplementary Table 1).

In addition to running *PRS-CSx* with multiple combinations of GWAS summary statistics, we also ran *PRS-CSx* with a range of global shrinkage parameter values ($\phi$). We applied a small range of $\phi$ values to each combination of GWAS summary statistics, (e.g., $\phi = 1e{-}6, 1e{-}4, 1e{-}2, 1$) to generate multiple panels of variant effect sizes for a single combination of GWAS summary statistics. This resulted in 152 total panels of variant effect sizes by applying 38 different combinations of GWAS summary statistics. Throughout the application of *PRS-CSx* we utilized LD panels that were made available in the GitHub repository https://github.com/getian107/PRScsx.

### Applying PRSs to cohort genetic data with *PREDICT*

For each of the 152 panels of variant effect sizes generated by *DISCOVER* we used Allelica's *PREDICT* software to compute individual-level PRS scores. The purpose of the *PREDICT* software is to take in a PRS panel (list of variants, effect alleles and effect sizes) and genotypes then output a PRS value for each individual who input a genotype. In other circumstances the *PREDICT* software can also calculate genetic risk percentiles and longitudinal risk projections. In this investigation we applied *PREDICT* to 12,751 individuals in the UK Biobank (all of which were not utilized in a previous GWAS), 5748 individuals within MESA, and 10,888 individuals within ARIC. After each PRS was computed it underwent ancestry adjustment and normalization. Ancestry adjustment was accomplished following an established approach[31] which subtracts the effect of the first four principal components from the PRS values (as indicated in equations 1 and 2 below). Ancestry adjustment was performed on the ancestry-specific cohorts separately. Normalization resulted in the PRSs having a mean of zero and standard deviation of one.

$$adjust\_model : PRS_{raw} \sim PC1 + PC2 + PC3 + PC4 \qquad (1)$$

$$PRS_{adjusted} = PRS_{raw} - adjust\_model(PC1, PC2, PC3, PC4) \qquad (2)$$

### PRS validation and testing

**Datasets**. As described above, we compiled a joint dataset of 29,387 individuals for the purpose of assessing the predictive value of the PRSs (Supplementary Tables 2 and 3). This joint dataset comprised the UK Biobank[23] (specifically the portion not already utilized for the GWAS), the MultiEthnic Study of Atherosclerosis[21] (MESA) and the Atherosclerosis Risk in Communities Study[22] (ARIC). Details of the collection of these datasets are detailed in the original publications. Briefly, the UK Biobank is a large, general purpose, prospective biobank containing approximately 500,000 individuals who were aged 40–69 when they were enrolled from 2006 and 2010. If a UK Biobank individual met any of the conditions listed in Supplementary Table 14, we defined the individual as being a CAD case. MESA is a diverse, moderate sized, atherosclerosis-focused study that contains approximately 6500 individuals who were aged 45?84 years when they were enrolled around 2000. If a MESA individual had a physiological event that was recorded as myocardial infarction, resuscitated cardiac arrest, definite angina, probable angina or coronary heart disease (CHD) death we denoted them as being a CAD case. ARIC is a prospective study of atherosclerosis that contains approximately 15,000 individuals enrolled from 1987 to 1998 who were aged 35–74 years. If an ARIC individual reported CAD at baseline or during a follow-up interview then we defined them as being a CAD case. Additional covariates such as age, sex, and family history of CAD were directly accessed from each cohort's descriptive data.

We applied quality control to each of these three datasets. First, we removed individuals from the UK Biobank dataset who were outliers in heterozygosity, contained putative sex chromosome aneuploidy, corresponded to a genotype missing rate greater than 10% or had excess relatives. All of these steps follow quality control flags provided by the UK Biobank[23], and have been taken by other investigators[50–52]. Second, in both ARIC and MESA we removed individuals whose (pre-imputation) genotype missing rate was greater than 10%. Third, the following variant level QC was applied to all three datasets: variants that had an allele frequency less than 0.01 or a Hardy–Weinberg Equilibrium $p$-value less than 1e−50 were removed. These quality control metrics fall in line with those others have applied[53–55].

**Genetic ancestry inference**. We inferred the ancestry of every individual in the Validation and Testing datasets with *iAdmix*[56] following default settings and with individuals from the 1000 Genomes[47] study as a reference dataset. For each individual we estimated the proportion of their genetic ancestry that originated from each of the 5 established One Thousand Genome Project continental-level superpopulations: African, American, East Asian, European and South Asian (Supplementary Table 15). These individual-level inferred proportions, which we also call ancestry components, are used in both the validation and testing of the polygenic risk scores in this investigation. Visualizations showing the first and second genetic principal component colored by cohort, reported ethnicity and the genetic ancestry defined above are provided in Supplementary Fig. 5.

**Data organization**. We used separate independent Validation and Testing datasets to assess the performance of the PRSs across different genetic ancestries (Supplementary Table 3). We split the entirety of the available data into these independent validation and testing datasets by following three straightforward guidelines: (1) keep cohorts together and (2) achieve at least 50 cases within the testing dataset and 10 cases within the validation dataset for each ancestry group and (3) maintain ancestry groups that are as homogeneous (non-admixed) as possible. Additionally, because of different goals corresponding to the validation and testing datasets we counted cases and analyzed the data within each differently. For the validation dataset we fit models with all individuals regardless of their ancestry, weighting their contribution to the model by their *iAdmix* ancestry component. For the testing dataset we fit models with only the individuals who belonged to the specific ancestry group. We followed this procedure in order to have the greatest statistical power as possible within the validation dataset and

have accuracy estimates that most faithfully matched the ancestry label in the testing dataset. As the models fit with validation datasets considered far more individuals than just those in a single ancestry group we lowered the number of cases needed for a single ancestry group. However, to maintain clarity the counts within Supplementary Table 3 for the validation dataset are based on single ancestry group thresholds.

We applied data splitting guidelines following the flow chart in Supplementary Fig. 2. For example, when considering the African ancestry group we counted 336 cases when placing the ARIC cohort in the validation dataset and 51 cases when placing the UK Biobank and MESA cohorts in the testing dataset. Another example, when considering the South Asian cohort nearly all of the cases were held in a single cohort, the UK Biobank. However, we could split this single cohort evenly and reach our guideline of having 50 cases in the testing dataset and 10 cases in the validation dataset. A final example, when considering the American ancestry group, no single cohort could be split evenly to reach the testing and validation dataset case quotas. Even when putting all of one cohort's cases in the testing dataset and splitting the other cohort cases across testing and validation datasets we were still unable to reach the case quotas. We therefore lowered the *iAdmix* ancestry component threshold used to count cases to 50%, and were then able to satisfy the case quotas.

The limited number of cases and controls within non-European ancestry groups increased the chance of generating an anomalous result. We attempted to prevent this by carefully constructing entirely independent validation and testing datasets, applying cross-validation repeatedly within the validation dataset (as will be described in the next section) and ultimately reporting the confidence intervals of all statistics generated from the testing dataset after appropriate control for multiple testing. The only way for us, and many others who have presented similar sample sizes in comparable genetic-based investigations[57–59], to achieve an exceedingly large sample size would be to limit the investigation to individuals of European ancestry, a move that would only increase the current inequity in healthcare.

**PRS validation.** Within each ancestry group of the validation dataset we aimed to identify a best performing PRS that could be evaluated in the testing dataset. We began with 152 PRS values for each individual. Each PRS value was generated from an unique mix of component GWASs and computational parameters as outline above. Following previous work[60,61] we theorized that rather than selecting a single PRS value for an individual it might be advantageous to form a single ensemble or meta PRS value. Our motivation was that each different combination of component GWASs and computational parameters captures a different aspect of the true underlying genetic architecture. For example, with one parameterization we may highlight rare high effect variants, in the same way that the clumping method might, whereas in another parameterization we may evenly weight thousands of marginally relevant variants, in the same way that the LDpred method might. By allowing the data, in a [meta PRS] cross-validation procedure, to determine which constituent PRSs from those generated assuming a range of genetic architectures, or methodological philosophies, best explain the data we may achieve the best possible risk predictions. This approach has been used by multiple previous publications[61–63]. However, we can not rule out that more predictive PRS panels may be generated with the use of additional methodologies in the future.

To form this meta PRS we applied elastic net logistic regression within a repeated cross-validation framework. Across 10 repeats and 10 folds we split the validation dataset into training and testing partitions. In both partitions the model included independent variables of age, sex, the top four genetic principal components, family history of CAD, cohort and all 152 polygenic risk scores. Specifically, on the training partition we determined the optimal $\lambda$

value of the elastic net logistic regression and on the testing partition we fit an elastic net logistic regression model with the previously determined $\lambda$ value. From this model fit to the testing partition we measured the coefficients of the polygenic risk score terms. At the end of the repeated, cross-validation process we generated 100 coefficients for each of the 152 PRSs, with each coefficient representing how salient or important the PRS was to the determination of the CAD case status. The average of the 100 coefficients became the final PRS weight used in the calculation of the meta PRS. The elastic net logistic regression was accomplished with *glmnet* and *cv.glmnet* functions from the *elasticnet* package within *R*. The repeated cross-validation process was organized with the *trainControl* function of the *caret* package within *R*.

While several other groups have either created metaPRSs by forming a weighted average of individual, predictive PRSs, or applied PRS-CSx upon a variety of multi-ancestry GWAS summary statistics, to the best of our knowledge no one else has previously combined both methodologies together. By doing so here, we present a novel way to generate a potentially maximally predictive PRS.

As described previously, to utilize the maximum amount of genetic information as possible (and thereby increase statistical power) we weighted the individuals in the *glmnet* models by their ancestry component. For example, when forming an African ancestry-specific meta PRS we would set the weights term in the model function call to the African ancestry component. We were then able to extend the number of individuals included in the model from those in just the ancestry group of the validation dataset to everyone in any ancestry-specific group of validation dataset.

Despite increasing the number of individuals within the meta PRS process, we found that the meta PRS could generate very uncertain risk predictions. To specify the degree of uncertainty, within each of the 100 total folds of the meta PRS generation process we formed a meta PRS from the coefficients estimated for the 152 PRS terms within that fold. We then regressed the scaled meta PRS against the phenotype and adjusted for the non PRS covariates of the glmnet model and lastly recorded the standard error of the meta PRS term. At the end of the meta PRS generation process we then had 100 standard error terms for each ancestry group. We decided that if the mean of the standard error terms was greater than 1 the meta PRS could not be relied upon as a precise risk predictor. To find an alternative PRS to apply within the testing dataset we utilized a bootstrapping procedure which subsampled the validation dataset, with replacement, 1000 times and fit a model for each of the 152 PRSs in each subsample. The PRS with the greatest average ORxSD was chosen as the single PRS to use for that ancestry group within the testing dataset.

All models computed upon the validation data included the covariates of the 152 PRSs, age, sex, top four genetic principal components, family history of CAD and cohort. Each of these covariates was normalized to a mean of zero and standard deviation of one. Any missing values were mean imputed. Each model was weighted by the *iAdmix* ancestry component corresponding to the group of individuals currently under analysis. Additionally, each PRS was ancestry adjusted by predictions made from a model fit to the top ten genetic principal components. This ancestry adjustment process is equivalent to that introduced by Hao et al.[31], and provides an additional means of correcting for fine-scale ancestry confounding of the polygenic risk scores. The average coefficients resulting from this process, which were then used to calculate the meta PRS, are provided within Supplementary Table 6 for all scores.

**PRS testing.** Within each ancestry group of the testing dataset we aimed to fairly measure the discriminative ability of the polygenic risk score derived within the validation dataset. Within each of the finalized ancestry groups of the testing dataset we created the meta PRS score by combining the raw PRS values with each of the weights derived from

the validation process. However, before combining the weights with the testing dataset PRSs, we first performed ancestry adjustment and normalization, following the same series of steps applied within the validation dataset. Alternatively, for ancestry groups that were not amenable to the metaPRS process we directly identified the PRS which performed best in a validation dataset bootstrapping process to serve as the ancestry-specific PRS in the testing dataset. We repeated this process for each set of ancestry-specific group for all individuals in the testing dataset. Therefore, each individual in the testing dataset had five, ancestry-specific polygenic risk scores.

These testing dataset ancestry-specific polygenic-risk scores were derived in the entirely independent validation datasets, meaning that any score that worked well in the validation dataset just by chance would almost certainly not also work well in the testing dataset. Furthermore, for most of the ancestry groups we utilized not just different individuals, but individuals from different cohorts between the validation and testing datasets, reducing the chance that some cohort-specific confounding may limit the ability of our scores to perform well in other cohorts in the future.

Next, within each cohort of each ancestry-specific group we formed a logistic regression model of the corresponding ancestry-specific PRS against CAD disease status adjusted for age, sex, top four genetic principal components, and family history. All of the adjusting covariates were normalized to a mean of zero and standard deviation of one. Following other authors, we utilized the top four genetic principal components when using these covariates in different analyses[64–67]. Furthermore, we conducted some sensitivity analyses and found that four genetic principal components were sufficient to predict the genetic ancestry assignments, within ten fold cross validation of the testing dataset, with an accuracy of 99%. The plots of all four genetic principal components graphically how certain combinations of components well split genetic ancestries (Supplementary Fig. 6).

The exponentiated coefficient of the PRS term in the model became the reported Odds Ratio per Standard Deviation (ORxSD). We also calculated the Brier score, Area Under the receiver operator Curve, Nagelkere $R^2$ and threshold at which a portion of the sample was at a certain multiple of odds compared to the remainder of the sample. When multiple cohorts were available for a single ancestry group (Africans, East Asians and Europeans) we performed a fixed effects meta-analysis with the meta package within the R computing language. We chiefly analyze the results of this meta-analysis, and not the results of the component cohorts, for these ancestry groups where a meta-analysis was conducted (Supplementary Table 16).

Individuals who we were unable to assign a single ancestry group, because they did not meet the *iAdmix* cut-off value of any group, were placed within a group we considered to be of admixed individuals. For these individuals we calculated an ancestry weighted PRS. This ancestry weighted PRS is a sum of the product between an individual's ancestry-specific PRS and the log(ORxSD) of that PRS measured within the corresponding single ancestry group and the corresponding ancestry component[68]. So if an individual was determined to be admixed with 20% European and 80% African ancestry their weighted PRS would be 0.2 of their European specific PRS value multiplied by log(1.56) plus 0.8 of their African specific PRS value multiplied by log(1.53). We additionally performed ancestry adjustment and normalization upon this weighted PRS within each ancestry group. For each ancestry group we thereby generated an admixture-aware PRS with a mean of zero and standard deviation of one. We then applied each of the models fit upon the single ancestry groups to the group of individuals with admixed ancestry to form a set of ancestry-specific predictions. Although, in this prediction calculation, the standard PRS was replaced with the weighted PRS. We additionally summed the ancestry prediction weighted by each individual's ancestry component to form a final,

single set of predictions. The logistic transformed predictions were compared to the true disease status of the individuals to calculate brier scores.

**Estimating the proportion of a population at twofold risk.** An additional statistic that measures the discriminative ability of a polygenic risk score is the odds against the remainder. The odds against the remainder is calculated by splitting a group of individuals according to whether their polygenic risk score is above or below a given threshold value, then calculating the odds ratio of disease of the high risk group against the low risk group. This type of odds ratio is often calculated at a specific quantile of the polygenic risk score within the group of individuals. Alternatively, we can work backwards and attempt to measure the quantile that achieves a specific odds against the remainder. And just as the odds ratio has a corresponding error range, the quantile value also has an error range. The single, point estimate of the quantile value is not the absolute two fold threshold but rather the best guess within a range of likely values. This idea builds upon Aragam et al.[11], who estimated the odds ratio conferred by being in the top quintile against the remainder of the distribution.

Just as a measured odds ratio is only relevant to the polygenic risk score it was computed from, the threshold is only relevant to a specific polygenic risk score applied to a specific testing population. Therefore, if the metaPRS panel we developed for a specific ancestry group generates a threshold of 80%, then only a PRS value calculated from the same panel upon the same ancestry group can be used with this 80% threshold.

The relatively small samples available to us made it unfeasible for us to calculate the odds threshold directly. Rather, we utilized a bootstrapping process which repeatedly sampled with replacement from an ancestry group within the testing dataset. With this boot-strapped sample we measured the odds against the remainder at a range of quantile values. Where possible (the African, East Asian and European ancestry groups) the final odds against the remainder value was calculated with a fixed effects meta-analysis (specifically with the metabin function of the meta package within the R computing language). After the bootstrap process we had a sample of odds against the remainder values at a variety of polygenic risk score quantiles. We were then able to fit a trend to the data and determine the exact polygenic risk score quantile at which the trend surpassed the desired odds against the remainder. In this fashion we computed the value recorded as the 2X threshold percentile.

**Estimating 10-year ASCVD risk with the Pooled Cohort Equations.** To estimate 10-year clinical risk using the Pooled Cohort Equations (PCE) we extracted and recoded information about clinical covariates from data available in the UK Biobank and MESA cohorts. This included: age, sex, smoking, LDL, HDL and total cholesterol, treated and untreated systolic blood pressure, diabetes status, family history of heart disease, CAD outcome.

Individuals within the Testing cohort were combined and filtered for adherence with 2018 American College of Cardiology (ACC)/American Heart Association (AHA) guidelines using the following Inclusion criteria: LDL comprised between 70 and lower than 190 mg/dL; Age at baseline comprised between 40 and 75; No Diabetes status at Baseline; No LDL-lowering medications at baseline; No missing values at any PCE variables. Re-calibrated 10-year risk was calculated for the PCE testing dataset as per Elliot et al.[69]. We estimated the baseline survival function in the PCE testing dataset through a Cox regression proportional hazard model (with PCE risk factors as control covariates) and combined it with published PCE Hazard Ratio values.

Comparison of risk category frequencies (Low, Borderline, Intermediate, and High 10-year PCE risk) between Not elevated and High genetic risk (genetic risk lower or equal/higher than twofold,

respectively) was assessed and no significant difference between risk frequencies in Low and High PRS groups was found (one-way $\chi^2$ test; $P$ = 0.983).

Finally, we performed separate confirmatory analyses using logistic regression of CHD and ASCVD as outcomes in the full PCE dataset recomputing the effect sizes (odds ratios) for all PCE covariates (age, sex, smoking, LDL, HDL and total cholesterol, treated and untreated systolic blood pressure, diabetes status, family history of heart disease) and a binary indicator variable denoting high CAD PRS (>2X threshold).

**Integrating CAD PRS into the PCE**. We used Allelica's *INTEGRATEpce* tool to assess the effect of integrating our ancestry-specific PRSs into the PCE model. The *INTEGRATEpce* software generally combines genetic risk predictions with non-genetic risk factors (such as cholesterol and medication status) in order to form a single, integrated disease risk prediction. In this investigation, we incorporated PRS-specific weights as the logarithm of the PRS hazard ratio (i.e. Log(HR)) with clinical risk weighted from the PCE model to output a single 10-year risk value. HRs for the PRSs were calculated in a Cox proportional hazards regression, with time of follow-up time variable and CAD as outcome. Cox regression was adjusted for age and first four Principal Components of ancestry, and used CAD as clinical outcome and time of follow-up as time variable. Cox regression was performed in the PCE testing dataset ($N$ = 9114) and because separate HRs are available for sex and ethnicity, we stratified by sex and self-reported ethnicity (African Americans and non-African Americans). Of note, the African American group comprised all individuals with African major genetic ancestry fraction, while the non-African Americans comprised individuals with European, South American, East-, and South-Asian major genetic ancestry fractions. As before, we analyzed the Net Reclassification Improvement (NRI) of PRS, considering two NRI measures: categorical NRI (calculated for a risk threshold of 20%) and continuous NRI (Supplementary Table 13).

#### Computational considerations
We analyzed the PRS with the coding languages R and Python. With each of these languages multiple packages/libraries were utilized. These packages/libraries are listed within Supplementary Table 17. Finally, as we have utilized multiple abbreviations within this text we have compiled a glossary within Supplementary Table 18 for easier reading.

#### Statistical considerations
When not otherwise noted the ranges around a point statistic are 95% confidence intervals. $P$-value significance thresholds were obtained after Bonferonni Correction.

#### Reporting summary
Further information on research design is available in the Nature Portfolio Reporting Summary linked to this article.

### Data availability
The UK Biobank is available to qualified researchers through https://www.ukbiobank.ac.uk/enable-your-research/apply-for-access. Both the MESA and ARIC datasets are freely available through dbGaP with accession codes phs000209.v2.p1 and phs000280.v3.p1, respectively. The 1000 Genomes dataset is freely available from multiple sources, including https://www.internationalgenome.org/. The polygenic risk scores described in this study are available under restricted access for non-commercial research use and can be obtained upon written request detailing their proposed use to the corresponding authors (giordano@allelica.com/george@allelica.com).

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

## Acknowledgements

We thank all individuals who contributed data to the publicly available datasets utilized in this study. UK Biobank data was accessed under application ID 40692. This study was funded by Allelica Inc.

## Author contributions

Concept and design: G.B.B., S.K., P.D., and G.B. Acquisition, analysis or interpretation of data: G.B.B., S.K., A.B., J.K., P.D., and G.B. Drafting of the manuscript: G.B.B. and S.K. Critical revision of the manuscript for important intellectual content: G.B.

## Competing interests

All authors are employees of Allelica, Inc.
