## [Peer Review File · Nature Communications]

Ancestry-Specific Polygenic Risk Scores are Risk Enhancers for Clinical Cardiovascular Disease AssessmentsREVIEWER COMMENTS

Reviewer #2 (Remarks to the Author):

This manuscript presents the development of a collection of polygenic risk scores (PRS) for atherosclerotic cardiovascular disease (ASCVD) with the aim of improving performance across diverse genetic ancestry groups. The authors utilized the PRS-CSx method to create ancestry-specific PRSs and explored various combinations of summary statistics to optimize performance. The resulting PRSs demonstrated similar levels of performance across all ancestry groups tested, with slight improvements over existing methods. While the findings are significant, there are several major concerns that need to be addressed before considering publication:

Major concerns:

1. The authors primarily applied an existing PRS method, PRS-CSx, to existing public datasets. While this work is necessary, it does not introduce highly innovative aspects on its own.
2. The validation sample sizes are exceptionally small and imbalanced across different ancestry groups. Considering the extensive number of combinations tested, there is a possibility that the observed results occurred by chance. Statistical testing with appropriate multiple-test correction should be conducted to validate the findings. Additionally, the manuscript should discuss the statistical power of the experiment and provide evidence based on existing data.
3. The PRSs developed in this study still fall short of SNP heritability and twin-based heritability estimates for ASCVD. Moreover, the performance of the PRSs is not significantly better than the existing approaches tested, as evidenced by overlapping confidence intervals.
4. The functionality and operation of Allelica's software tools (DISCOVER, PREDICT, and INTEGRATEpce) described in the manuscript lack clarity and may cause confusion when compared to existing approaches. It is recommended to provide supplementary materials or clarify the pipeline of these tools to enhance understanding.
5. A more detailed comparison among the different PRSs generated in the study should be included to assess the factors contributing to improved performance in different ancestry groups. It is crucial to examine the extent of improvement achieved by the meta PRS over other approaches, particularly in ancestries where it outperforms the rest. Additionally, more comparisons between covariate-only models and the best-performing PRS models in Table 1 would help evaluate the performance of the multi-ancestry PRS across different groups.
6. The manuscript and supplementary materials lack essential details concerning genotype data quality control (QC) and other relevant specifications that should be reported in multi-ancestry analyses. Comprehensive descriptions of sample-level QC, imputation, and ancestry-specific QC information for the UK Biobank (UKBB), Multi-Ethnic Study of Atherosclerosis (MESA), and Atherosclerosis Risk in Communities (ARIC) cohorts should be provided to address this concern.
7. The authors propose CAD risk thresholds based on PRS for each ancestry and subsequently suggest a Risk Assessment Tool. It is important to justify why PRS-CSx was exclusively adopted for this analysis. While PRS-CSx is a well-regarded methodology, it is essential to consider the uncertainty associated with PRS and account for differences among various methods when establishing criteria or numerical values for risk stratification.
8. The definition of CAD cases and the prevalence of CAD in the UK Biobank (UKBB) cohort (26,135/433,131, ~6%) differ from other studies. The manuscript should clarify the phenotype definitions used in comparison to previous studies to address this discrepancy.
9. Recent studies have reported inconsistencies in rank concordance and uncertainty among different PRS methodologies, indicating the insufficiency of relying solely on PRS scores derived from a single methodology as indicators for individuals. The manuscript should discuss these findings and their implications.
10. It is unclear whether the risk thresholds for each racial group can be consistently observed across different PRS methods. The rationale behind exclusively using PRS-CSx to represent risk levels should be elucidated.
11. To gain a better understanding of the optimal use of PRS, the manuscript should include additional

analyses and discussions, which would help suggest future directions in the field.

12. The use of principal components (PCs) is inconsistent, with some instances using 10 PCs and others using 4. Given that PCs are derived using the 1000 Genomes Project, a minimum of 10 PCs is necessary to distinguish individuals with diverse genetic ancestries. A sensitivity analysis involving additional PCs or other reasonable techniques is required.

13. The authors employed POLYFUN for fine-mapping in CARDIOGRAMplusC4D without clearly elucidating the insights or impact of this specific component in their experiments. The manuscript should explain why only CARDIOGRAMplusC4D was analyzed and clarify its significance within the context of the authors' experiments.

14. If approximately 450,000 EUR samples were used in the internally computed GWAS analyses, it is anticipated that the authors controlled for relatedness inflation by adopting the fastGWA-GLMM model at the GWAS analysis level. However, it is unclear whether explicit adjustment for relatedness was performed in the genetic data QC step.

15. While the GWAS analysis approach seems reasonable, there may be concerns regarding the validation and establishment of a target set for PRS. It would be helpful if the authors could provide insights on familial relationships or relatedness among the samples used in both the GWAS and PRS analyses.

16. Potential issues of sample duplication or familial relationships should be identified and at least acknowledged as limitations in the discussion. Such issues can lead to inflationary effects on performance.

Minor concerns:

1. Typos in the manuscript need to be corrected, particularly the inconsistent use of 'CARDIOGRAMplusC4D.'

2. LD panels for each ancestry used in PRS-CSx should be provided.

3. In Table S3, the definition of the "American" group in the dataset and the determination of the continent in UKBB data should be clarified. Considering the limited number of samples, it is worth reconsidering the necessity of this category.

4. The manuscript does not clearly explain why only CardioGRAM was fine-mapped while other cohorts were not. The study design regarding fine mapping should be clarified.

5. Figure S1 states "The created 152 PRS values for 15,664 total individuals." The reasons behind selecting 15,664 samples and the discrepancy in sample sizes between the validation and test datasets (14,476 and 7,964, respectively) should be addressed, especially considering that the total sample size for UKBB (test) + MESA + ARIC is 22,440. If some samples were discarded, the rationale should be explained.

6. The term "AFR-2" is used for GWAS employed in various instances (Table S7 and others), but it is not defined in the manuscript.

Reviewer #3 (Remarks to the Author):

The manuscript from Busby et al presents an analysis of a CAD PRS as a risk enhancer of the current PCE risk model for ASCVD. They generate new CAD PRSs from multi-ancestry GWAS summary statistics then perform validation and downstream risk reclassification analyses in multiple ancestries of UK Biobank and MESA. Overall, the manuscript looks well designed, analysed and interpreted. The findings are impactful and would add substantially to the literature around the quantification of the clinical utility of a CAD PRS. It is particularly timely as multiple major cardiology bodies around the world are assessing their risk prediction/prevention guidelines and PRSs are being considered as risk enhancers (or for a similar purpose).

I have only minor comments, largely concerning clarity and presentation.

- The presentation of the results could be improved. In particular, Table 1 should be a multi-panel

figure presenting the cross-ancestry performance of various metrics.

- Table S4 should be a main table as a full page of results (pg4) is referencing the information in it. Table S4 is also quite comprehensive and carries the information in Tables S5 and S6.

- In estimating the proportion of a population at 2x risk, it would be more useful to have a confidence interval on the % threshold. If I understand correctly, this should be straightforward to calculate from the existing bootstrap procedure.

- In comparing the %thresholds between PRSs, it's unclear how the authors arrive at avg differences of 7.0% and SD 3.74%. A 3.74% difference in SD at these thresholds (~85% from Table 1) does not correspond to a difference of 7% in terms of the area under the normal curve (it's closer to 0.7%). Please clarify.

- Tables S12 and S13 should have both UKB and MESA columns as well as them combined

- The authors would be encouraged to make the CAD PRS they develop freely available (i.e. variants and weights) as per the PRS Reporting Standards

Response to Reviewer's Comments

Throughout, we have put the **Reviewer's comments in bold text** with our response in normal weighted text.

Reviewer #2 (Remarks to the Author):

This manuscript presents the development of a collection of polygenic risk scores (PRS) for atherosclerotic cardiovascular disease (ASCVD) with the aim of improving performance across diverse genetic ancestry groups. The authors utilized the PRS-CSx method to create ancestry-specific PRSs and explored various combinations of summary statistics to optimize performance. The resulting PRSs demonstrated similar levels of performance across all ancestry groups tested, with slight improvements over existing methods. While the findings are significant, there are several major concerns that need to be addressed before considering publication:

Major concerns:

1. The authors primarily applied an existing PRS method, PRS-CSx, to existing public datasets. While this work is necessary, it does not introduce highly innovative aspects on its own.

We understand how the use of a single, existing PRS methodology may give the appearance of a methodology that is not highly innovative. However, we believe that the current landscape of multi-ancestry PRS literature, our specific application of the tool, and the larger direction of our investigation would suggest that our methodology is innovative.

The PRS literature cites the original, non-multi-ancestry PRS-CS method four times as often as the multi-ancestry PRS-CSx method. While many other non-multi-ancestry methods, such as LDpred and lassosum, are heavily cited, there are few other well known multi-ancestry methods. Therefore, the use of PRS-CSx is relatively unique in terms of the entirety of PRS literature. Even so, we did not just apply PRS-CSx following the standard process mentioned in the documentation. Rather we applied PRS-CSx over multiple different parameterizations then created a meta-PRS from the many outputs. While meta-PRSs, or PRSs that are created through the combination of other existing PRSs, are occasionally described in the literature, we are not aware of an example where a meta-PRS is made from PRS-CSx outputs. Lastly, we utilize the meta-PRS of PRS-CSx outputs not just within a basic regression analysis, but rather in a complex clinical model. While other researchers do infrequently extend their own PRS investigation to complex clinical models, we are unaware of an investigation that also uses a multi-ancestry PRS framework.

To better highlight the multi-faceted manner in which our investigation is novel, with specific focus on the use of PRS-CSx, we have changed the text in several locations of our manuscript.

Line 85

To the best of our knowledge, the output of PRS-CSx has never been previously used to create a metaPRS in this way

Line 217

In the context of this study, this means utilizing independently validated ancestry-specific PRSs that are able to identify individuals with higher genetic risk.

Line 429

While several other groups have either created metaPRSs by forming a weighted average of individual, predictive PRSs, or applied PRS-CSx upon a variety of multi-ancestry GWAS summary

statistics, to the best of our knowledge no one else has previously combined both methodologies together. By doing so here we present a novel way to generate a potentially maximally predictive PRS.

2. The validation sample sizes are exceptionally small and imbalanced across different ancestry groups. Considering the extensive number of combinations tested, there is a possibility that the observed results occurred by chance. Statistical testing with appropriate multiple-test correction should be conducted to validate the findings. Additionally, the manuscript should discuss the statistical power of the experiment and provide evidence based on existing data.

We readily agree that our sample sizes for the non-European groups are unfortunately smaller than desired. Thanks to this comment we now realize that our original manuscript both did not take advantage of ways in which we should have remedied this problem and we did not properly describe other ways in which we did attempt to remedy this problem.

In response to this comment, we performed three additional analyses / text updates:

1. We conducted multiple hypothesis correction on all the P-values that we report, allowing the values to be more easily interpreted.
2. We have added text that both clarifies how our sample sizes were relatively small and how these sample sizes may have impacted the reported results.
3. We have added further text that we believe better explains the steps we took during the validation process in order to reduce the chance of overfitting and anomalous positive results.

Line 594

P-value significance thresholds were obtained after Bonferonni Correction.

Line 389

The limited number of cases and controls within non-European ancestry groups increased the chance of generating an anomalous result. We attempted to prevent this by carefully constructing entirely independent validation and testing datasets, applying cross-validation repeatedly within the validation dataset (as will be described in the next section) and ultimately reporting the confidence intervals of all statistics generated from the testing dataset after appropriate control for multiple testing.

Line 476

These testing dataset ancestry-specific polygenic-risk scores were derived in entirely independent validation datasets, meaning that any score that worked well in the validation dataset just by chance would almost certainly not also work well in the testing dataset. Furthermore, for most of the ancestry groups we utilized not just different individuals, but individuals from different cohorts between the validation and testing datasets, reducing the chance that some cohort-specific confounding may limit the ability of our scores to perform well in other cohorts in the future.

3. The PRSs developed in this study still fall short of SNP heritability and twin-based heritability estimates for ASCVD. Moreover, the performance of the PRSs is not significantly better than the existing approaches tested, as evidenced by overlapping confidence intervals.

Thanks to this comment we now realize that our original manuscript did not have enough background on the structure and practical uses of polygenic risk scores. To rectify this situation we have edited text to describe how SNP heritability is the upper limit of polygenic risk score accuracy, and how several studies have shown that this upper limit can be hit once studies are completed with very massive sample sizes and comprehensive genotyping. Therefore, we are not alarmed that the PRSs

we developed fall short of SNP heritability. Furthermore, we now explain how we compare the performance of one PRS to another. In short, if one PRS shows greater accuracy than another PRS across a variety of statistics we believe that it should be chosen for future use. An analogy is that if machine A leads to a greater profit than machine B, then machine A would be installed in a new factory even if the difference in profit is somewhat small. Choosing machine B, or in our case the less accurate score, would opt for a worse desirable outcome in light of the evidence at hand. We now concisely communicate this notion in the text.

Line 275

The variance explained by the resultant score can theoretically reach the SNP-level of heritability. However, in practice factors such as limited sample size prevent the score from reaching even close to the heritability limit [48-50].

Line 102

These differences suggest that not utilizing the novel PRSs for future use would likely lead to relatively inferior predictions.

4. The functionality and operation of Allelica's software tools (DISCOVER, PREDICT, and INTEGRATEpce) described in the manuscript lack clarity and may cause confusion when compared to existing approaches. It is recommended to provide supplementary materials or clarify the pipeline of these tools to enhance understanding.

Following this comment we have inserted additional explanatory text to the manuscript which covers the general purpose and design of each of these pieces of software. We intend for these additions to adequately convey the pipeline the data takes through the software.

Line 283

We implemented PRS-CSx within Allelica's DISCOVER PRS development software. The purpose of the DISCOVER software is to take in summary statistics and output a panel of variants, alongside effect sizes, that can be used to create a polygenic risk score of superior accuracy. This design matches the design of other well known genetics software, like LDpred and PRS-CS. In this specific investigation, within `DISCOVER` we combined GWAS summary statistics that were generated from different ancestral groups so that the modified variant effects in the output would benefit from a diversity of data.

Line 570

We used Allelica's INTEGRATEpce tool to assess the effect of integrating our ancestry-specific PRSs into the PCE model. The INTEGRATEpce software generally combines genetic risk predictions with non-genetic risk factors (such as cholesterol and medication status) in order to form a single, integrated disease risk prediction. In this investigation, we incorporated PRS-specific weights as the logarithm of the PRS hazard ratio (i.e. $\text{Log}(\text{HR})$) with clinical risk weighted from the PCE model to output a single 10 year risk value.

Line 309

For each of the 152 panels of variant effect sizes generated by DISCOVER we used Allelica's PREDICT software to compute individual level PRS scores. The purpose of the PREDICT software is to take in a PRS panel (list of variants, effect alleles and effect sizes) and genotypes then output a PRS value for each individual who input a genotype. In other circumstances the PREDICT software can also calculate genetic risk percentiles and longitudinal risk projections.

5. A more detailed comparison among the different PRSs generated in the study should be included to assess the factors contributing to improved performance in different ancestry groups. It is crucial to examine the extent of improvement achieved by the meta PRS over other approaches, particularly in ancestries where it outperforms the rest. Additionally, more comparisons between covariate-only models and the best-performing PRS models in Table 1 would help evaluate the performance of the multi-ancestry PRS across different groups.

Following this comment we have created two new supplementary Figures (S5 and S6) that report the Odds Ratio per standard deviation and Area Under the receiver operating Curve for models comprising various sets of covariates. For example, in supplementary table S4 we reported the accuracy statistics of the PRS when the covariates of PC1-4, age, sex and family history were included in the larger model. The new Figures include the same variety of statistics but with different sets of covariates, such as just PC1-4 or just age.

Line 115

Additional statistics, such as Nagelkerke R^2 and the Area Under the receiver operator Curve, and models containing additional sets of covariates, such as age and/or sex alone, further confirmed that our novel multi-ancestry score performed better than the alternative scores (Figures 2, S5 and S6 and Tables S4 and S6).

6. The manuscript and supplementary materials lack essential details concerning genotype data quality control (QC) and other relevant specifications that should be reported in multi-ancestry analyses. Comprehensive descriptions of sample-level QC, imputation, and ancestry-specific QC information for the UK Biobank (UKBB), Multi-Ethnic Study of Atherosclerosis (MESA), and Atherosclerosis Risk in Communities (ARIC) cohorts should be provided to address this concern.

This comment identifies useful additional information that was not present in the original manuscript. We have added details on the quality control metrics that we previously applied, resolving the issue raised by this comment:

Line 341

We applied quality control to each of these three datasets. First, we removed individuals from the UK Biobank dataset who were outliers in heterozygosity, contained putative sex chromosome aneuploidy, corresponded to a genotype missing rate greater than 10% or had excess relatives. Second, in both ARIC and MESA we removed individuals whose (pre-imputation) genotype missing rate was greater than 10%. Third, the following variant level QC was applied to all three datasets: variants that had an allele frequency less than 0.01 or a Hardy-Weinberg Equilibrium p-value less than $1e-50$ were removed. These quality control metrics fall in line with those others have applied.

7. The authors propose CAD risk thresholds based on PRS for each ancestry and subsequently suggest a Risk Assessment Tool. It is important to justify why PRS-CSx was exclusively adopted for this analysis. While PRS-CSx is a well-regarded methodology, it is essential to consider the uncertainty associated with PRS and account for differences among various methods when establishing criteria or numerical values for risk stratification.

We exclusively utilized PRS-CSx as the GWAS summary statistic-to-PRS panel method for two primary reasons. First, at the time of writing there were few ancestry-aware methods that can be reliably implemented. Therefore, it seemed appropriate to only apply PRS-CSx. Evidence for this line of thinking is that multiple other publications only utilized the PRS-CSx method, including “Development and validation of a trans-ancestry polygenic risk score for type 2 diabetes in diverse populations” by Tian Ge et al. and “The construction of cross-population polygenic risk scores using

transfer learning” by Zhangchen Zhao et al. Second, we did not just create a single PRS with the PRS-CSx method, but rather over a hundred and then combined them together using a metaPRS approach. Under this approach, we are able to apply a wide variety of parameterizations that capture a range of genetic architectures, including an architecture of a few, highly important variants and an architecture of thousands of modestly important variants. The metaPRS approach is then able to determine which architecture, or mix of architectures, corresponds to the best predictions. We have added text to the manuscript to better explain this feature of the metaPRS approach.

This comment also led us to realize that we may have not adequately reported the range of error surrounding the genetic risk threshold. Specifically, the risk threshold is the percentile at which those at higher risk have twice the odds of developing CAD as those at lower risk. As this odds has an error bound we also have an error bound for the threshold. We report this threshold (including in the new Figure 2) and now better communicate the fact that threshold is not known for certain, but rather is our best guess within a likely range of thresholds.

Line 132

Similar to the ORxSD, this threshold carries a range of error that is exacerbated by the size of the sample analyzed. Nevertheless its point estimate is equivalent to that of well established risk factors such as family history and some Mendelian-inherited genetic variants

Line 404

Our motivation was that each different combination of component GWASs and computational parameters captures a different aspect of the true underlying genetic architecture. For example, with one parameterization we may highlight rare high effect variants, in the same way that the clumping method might, whereas in another parameterization we may evenly weight thousands of marginally relevant variants, in the same way that the LDpred method might. By allowing the data, in a cross validation procedure, to determine which type (or types) of genetic architecture, or methodological philosophy, is most appropriate to the dataset we may achieve the best possible risk predictions.

Line 525

And just as the odds ratio has a corresponding error range, the quantile value also has an error range. The single, point estimate of the quantile value is not the absolute two fold threshold but rather the best guess within a range of likely values.

8. The definition of CAD cases and the prevalence of CAD in the UK Biobank (UKBB) cohort (26,135/433,131, ~6%) differ from other studies. The manuscript should clarify the phenotype definitions used in comparison to previous studies to address this discrepancy.

We thank the reviewer for highlighting this potential discrepancy. Given the importance of phenotype definitions, we want to ensure that our numbers are correct. We therefore performed some additional checks on our data and numbers and have no reason to believe that they are inconsistent with previous work.

Previous publications have chosen different ways to identify individuals diagnosed with CAD, (for example, Khera et al 2018, Inouye et al 2018, Privè et al 2020, Bolli et al 2021). We choose the definitions provided by Inouye et al 2018, which - to the best of our knowledge - represent the most detailed and comprehensive definition formulated so far.

The UKBB field numbers and outcome codes we used are reported below:

1. Hospital Electronic Record: Primary and secondary Diagnoses - ICD10 codes (outcome and date Fields 41270 and 41280, respectively)
 - codes from I21 (I21.0) to I24 (I24.9) and I25.2
2. Hospital Electronic Record: Primary and secondary Diagnoses - ICD9 codes (outcome and date Fields 41271 and 41281, respectively)
 - codes 410 to 412
3. Hospital Electronic Record: Operative procedures - OPCS4 (outcome and date Fields. 41272 and 41282)
 - codes from K40 to K46, K49, K50.1, K50.2, K50.4 or K75
4. Non-cancer illness code, self-reported (outcome and interpolated year Fields 20002 and 20008, respectively)
 - code 1075
5. Self reported Operation code (outcome and interpolated year Fields 20004 and 20010, respectively)
 - codes 1070, 1095, or 1523
6. Self reported Age heart attack diagnosed (Field N. 3894 that is related with outcome Field 6150 and code 1, mentioned in Inouye et al 2018)
7. Underlying primary and secondary cause of death: ICD10 (Fields N. 40001 and N. 40002, respectively)
 - Same codes of point 1.

We calculated the total fraction of CAD cases in UKBB as: $N.cases / (N.Cases+N.controls)$ with values of N.Cases and N.Controls reported in Table S2 (i.e. “UK Biobank European”). The computed percentage of CAD cases in UKBB European is $26,178 / (26,178+433,995) = 5.7\%$ while Inouye reported a Total fraction of cases (occurred before the age of 75) of about 4.60%. As the Inouye paper was published in 2018, with a reported end of follow-up for Hospital Electronic records of March 2015, their numbers were based on fewer years of follow up compared to our analysis. We would therefore not expect the numbers to be perfectly aligned.

We further tested this assumption by selecting from our dataset all CAD cases that occurred before 01/03/2015 and with age of CAD <75, we obtained a total fraction of CAD cases of 5.0%. The residual small discrepancy of 0.4% of CAD cases is likely due to differences in the UKBB sub-samples used as well as in differences of end of follow-up for other UKBB fields (e.g. self-reported CAD cases).

The fraction of CAD cases observed in the UKBB Europeans was much closer to that observed in a later study by Privé et al 2019 (CAD: 5.42%) . In this case, Operative procedures (OPCS)4 related outcomes were not considered and so this CAD fraction is likely under-estimated compared to our analysis. Khera et al 2018 also reported a smaller CAD fraction (between 3.0 % and 3.4%) because of shorter UKBB follow-up and because only prevalent CAD cases were considered. Overall, the fraction of CAD cases identified in the UKBB appears to be consistent with older literature, given differences in UKBB follow-up length and CAD definitions.

Finally, we edited Table S11 to include additional information about UKBB fields and codes used to define CAD outcome and to aid future replicability. We have reproduced Table S11 below:

Outcome-defining UKBB Field	Time-defining UKBB Field	Event Code
Primary and secondary Diagnoses - ICD10 codes (41270)	41280	I21 to I24 or I25.2

Primary and secondary Diagnoses - ICD9 codes (41271)	41281	410 to 412
Operative procedures - OPCS4 (41272)	41282	K40 to K46, K49, K50.1, K50.2, K50.4 or K75
Non-cancer illness code, self-reported (20002)	20008	1075
Self reported Operation code (20004)	20010	1070, 1095, or 1523
Vascular/heart problems diagnosed by doctor (6150)	3894	1
Underlying primary and secondary cause of death: ICD10	40001 and 40002	I21 to I24 or I25.2

9. Recent studies have reported inconsistencies in rank concordance and uncertainty among different PRS methodologies, indicating the insufficiency of relying solely on PRS scores derived from a single methodology as indicators for individuals. The manuscript should discuss these findings and their implications.

We agree that applying multiple methodologies, not just a single method such as PRS-CSx, may increase the chances of developing a maximally accurate PRS. However, as we note in our responses to comments #1 and #7, we utilized a metaPRS approach that covers multiple types of genetic architectures in the same way that multiple methodologies do. Therefore, following the results of other studies that implemented a metaPRS, we believe that our approach is not necessarily insufficient. Nevertheless, to cover this aspect of the metaPRS approach more clearly we have added text to the manuscript that considers the effects of implementing a single methodology.

Line 409

By allowing the data, in a [meta PRS] cross validation procedure, to determine which constituent PRSs from those generated assuming a range of genetic architectures, or methodological philosophies, best explain the data we may achieve the best possible risk predictions. This approach has been used by multiple previous publications [62-64]. However, we can not rule out that more predictive PRS panels may be generated with the use of additional methodologies in the future.

10. It is unclear whether the risk thresholds for each racial group can be consistently observed across different PRS methods. The rationale behind exclusively using PRS-CSx to represent risk levels should be elucidated.

The risk thresholds we calculated are only applicable to the specific PRS panel that they were calculated from and are a function of both the effect sizes computed for the variants in that panel and the overall predictive performance (ORxSD) of the PRS panel in the testing population. For example, we estimate the threshold for our novel metaPRS to be 76% for the South Asian specific score in South Asian individuals, while the threshold for the GPS_CAD score specific to South Asian individuals was 77%. These threshold values are not tied to the PRS-CSx method, but rather the specific panel of variant weights that we used to generate the PRS. This specificity mimics how a method such as LDpred might produce an odds ratio of 1.2 when applied to a low powered GWAS with poor parameters and it may produce an odds ratio of 1.5 when applied to high powered GWAS with optimal parameters. While the same disease is being considered, the odds ratio changes because it is not tied to the method, but rather LDpred's output panel of variant effects. To clarify this point we have added the following text to the manuscript.

Line 530

Just as a measured odds ratio is only relevant to the polygenic risk score it was computed from, the threshold is only relevant to a specific polygenic risk score applied to a specific testing population. Therefore, if the metaPRS panel we developed for a specific ancestry group generates a threshold of 80%, then only a PRS value calculated from the same panel upon the same ancestry group can be used with this 80% threshold.

11. To gain a better understanding of the optimal use of PRS, the manuscript should include additional analyses and discussions, which would help suggest future directions in the field.

We totally agree that we originally focused our discussion on limitations and the present state of the risk prediction field without clearly articulating areas for future work. We have therefore added a sentence to the Discussion to include several points in which future individuals who study polygenic risk scores may apply their efforts.

Line 192

Future work that jointly considers the environmental and genetic effects of ancestry differences on risk predictions could also provide a potential avenue for further developing the approaches outlined here.

12. The use of principal components (PCs) is inconsistent, with some instances using 10 PCs and others using 4. Given that PCs are derived using the 1000 Genomes Project, a minimum of 10 PCs is necessary to distinguish individuals with diverse genetic ancestries. A sensitivity analysis involving additional PCs or other reasonable techniques is required.

Thankyou for identifying an inconsistency in the use of principal components. Throughout, we use the top four, not ten, genetic principal components. The text of the manuscript erroneously reported ten in a few places, an error that has now been fixed.

Whilst we thank the reviewer for the suggestion of using 10 PCS, we have decided to stay with four because many other publications report the use of only four genetic principal components, including the 2019 paper from Khera et al. that first described PC-based ancestry adjustment and which has become a foundation of many recent investigations. We also use continental level ancestry groups throughout and the first 4 PCs are sufficient to differentiate individuals within these groups. Nevertheless, to verify that only four principal components can be used to distinguish different continental populations we created a plot of all four principal components of the testing dataset and now include these in the manuscript (Figure S4). The plot shows how the first principal component identifies the African population, the second principal component identifies East Asian and European populations, the third principal component identifies the South Asian population and the fourth principal component largely identifies the American population. The use of a k-Nearest Neighbors “learned” these types of rules and was able to predict genetic ancestry with an accuracy greater than 99%. This high level of accuracy leads us to believe that four principal components are sufficient to both capture the ancestry of the individuals used and to distinguish individuals within each of these diverse ancestries.

Line 486

Following other authors, we utilized the top four genetic principal components when using these covariates in different analyses [65-68]. Furthermore, we conducted some sensitivity analyses and found that four genetic principal components were sufficient to predict the genetic ancestry assignments, within ten fold cross validation of the testing dataset, with an accuracy of 99%. The plots

of all four genetic principal components graphically how certain combinations of components well split genetic ancestries (Figure S4).

13. The authors employed POLYFUN for fine-mapping in CARDIOGRAMplusC4D without clearly elucidating the insights or impact of this specific component in their experiments. The manuscript should explain why only CARDIOGRAMplusC4D was analyzed and clarify its significance within the context of the authors' experiments.

We used POLYFUN to finemap the CARDIOGRAMplusC4D summary statistics in order to identify a set of putatively functional variants and weights. We restricted this analysis to only these summary statistics in order to add functionally informed data into the analysis whilst keeping the overall number of different GWASs to a manageable number. Whilst this GWAS was not the largest available in terms of absolute number of individuals, it contained the largest number of cases (>2 times more than our UKB GWAS) and so represented a good candidate GWAS for identifying putatively causal variants.

Line 253

We only applied fine-mapping to the CardioGRAMplusCD4 summary statistics because it allowed us to introduce functional information into our investigation with the best chance of utility, owing to the CardioGRAMplusCD4's great statistical power, without creating too many, unwieldy modified summary statistics.

14. If approximately 450,000 EUR samples were used in the internally computed GWAS analyses, it is anticipated that the authors controlled for relatedness inflation by adopting the fastGWA-GLMM model at the GWAS analysis level. However, it is unclear whether explicit adjustment for relatedness was performed in the genetic data QC step.

Please see our response to comment #6 on genetic data QC performed for this analysis, including the removal of related individuals from the UK Biobank. As we removed related individuals additional adjustment was unnecessary.

15. While the GWAS analysis approach seems reasonable, there may be concerns regarding the validation and establishment of a target set for PRS. It would be helpful if the authors could provide insights on familial relationships or relatedness among the samples used in both the GWAS and PRS analyses.

Within the UKB GWAS, we performed QC that removed individuals with excess number of relatives, which limits inflation issues caused by relatedness (see our response to comment #6 regarding QC performed on the datasets used in this study).

16. Potential issues of sample duplication or familial relationships should be identified and at least acknowledged as limitations in the discussion. Such issues can lead to inflationary effects on performance.

We can confirm that no duplicated individuals were present in the three cohorts and point the reviewer to our answers to points 6 and 15 where we describe the relatively standard approaches we have deployed for correcting for relatedness (in the datasets are not full of families). In the text of the manuscript we do point out that the individuals across the datasets are independent, which should assuage concerns regarding duplication across datasets.

Minor concerns:

1. Typos in the manuscript need to be corrected, particularly the inconsistent use of 'CARDIOGRAMplusC4D.'

We have addressed the inconsistent use of "CARDIOGRAMplusC4D" and other typos, thank you for highlighting their presence.

2. LD panels for each ancestry used in PRS-CSx should be provided.

We have updated the manuscript to indicate where we accessed the LD panels that were utilized when running PRS-CSx.

Line 305

Throughout the application of PRS-CSx we utilized LD panels that were made available in the GitHub repository <https://github.com/getian107/PRScsx>.

3. In Table S3, the definition of the "American" group in the dataset and the determination of the continent in UKBB data should be clarified. Considering the limited number of samples, it is worth reconsidering the necessity of this category.

We have provided a more extensive caption to Table S3 that covers the way we defined the "American" group. Specifically, we designated someone from the testing and validation dataset as American if they were genetically similar to the individuals in the 1000 Genomes Project whose continental/superpopulation was "American". This process is similarly completed for the European group by calculating genetic similarity to the individuals in the 1000 Genomes Project whose continental/superpopulation was "European". The iAdmix tool completes the calculations that generate the exact similarity measure. The caption now points out places in the text where further information regarding the ancestry inference can be found.

4. The manuscript does not clearly explain why only CardioGRAM was fine-mapped while other cohorts were not. The study design regarding fine mapping should be clarified.

We describe our reasoning for choosing to finemap the CARDIOGRAMplusCD4 in response to comment #13 above, alongside the respective change to the text.

5. Figure S1 states "The created 152 PRS values for 15,664 total individuals." The reasons behind selecting 15,664 samples and the discrepancy in sample sizes between the validation and test datasets (14,476 and 7,964, respectively) should be addressed, especially considering that the total sample size for UKBB (test) + MESA + ARIC is 22,440. If some samples were discarded, the rationale should be explained.

Thanks to this comment we now realize that in a previous iteration of this investigation we excluded some admixed individuals which shifted the testing data set size to the 7,946 value reported in Figure S1. We have now modified Figure S1 and harmonized all of the other sample size values so that they align to Table S3.

6. The term "AFR-2" is used for GWAS employed in various instances (Table S7 and others), but it is not defined in the manuscript.

We erroneously used the term "AFR-2" when meant to write "AFR-1". The manuscript now reflects this fact.

Reviewer #3 (Remarks to the Author):

The manuscript from Busby et al presents an analysis of a CAD PRS as a risk enhancer of the current PCE risk model for ASCVD. They generate new CAD PRSs from multi-ancestry GWAS summary statistics then perform validation and downstream risk reclassification analyses in multiple ancestries of UK Biobank and MESA. Overall, the manuscript looks well designed, analysed and interpreted. The findings are impactful and would add substantially to the literature around the quantification of the clinical utility of a CAD PRS. It is particularly timely as multiple major cardiology bodies around the world are assessing their risk prediction/prevention guidelines and PRSs are being considered as risk enhancers (or for a similar purpose).

I have only minor comments, largely concerning clarity and presentation.

- The presentation of the results could be improved. In particular, Table 1 should be a multi-panel figure presenting the cross-ancestry performance of various metrics.

- Table S4 should be a main table as a full page of results (pg4) is referencing the information in it. Table S4 is also quite comprehensive and carries the information in Tables S5 and S6.

Thank you for these suggestions. We have now added a forest plot reporting the ORxSD of the best performing scores to Figure 1 and have added a new Figure 2 that shows the benchmarking analysis previously shown in Table S4. We have removed Table 1 as this information is now captured across both figures.

- In estimating the proportion of a population at 2x risk, it would be more useful to have a confidence interval on the % threshold. If I understand correctly, this should be straightforward to calculate from the existing bootstrap procedure.

We have since calculated the confidence interval of the 2x percent threshold. These interval values are now available in Figure 2 and Table S5, and are referenced in the main text of the manuscript.

- In comparing the %thresholds between PRSs, it's unclear how the authors arrive at avg differences of 7.0% and SD 3.74%. A 3.74% difference in SD at these thresholds (~85% from Table 1) does not correspond to a difference of 7% in terms of the area under the normal curve (it's closer to 0.7%). Please clarify.

To calculate the 7.0% average difference we used the threshold values in Table S4. For example, with regards to the group of South Asian ancestry the threshold for our newly presented PRS was 76% and the lowest threshold of a competing PRS was 77%. The difference for the South Asian ancestry group would then be 1%. The difference for the African, East Asian, European, and American ancestry groups was 10%, 8%, 10% and 6%, respectively. The average across all of these difference values was 7.0%.

We have modified the manuscript to more clearly indicate that the values used to calculate this average came from Table S4. This change should allow others to recreate the calculation just described.

Line 152

the average difference between the two-fold threshold found with all of the Allelica_CAD_vJ scores and the lowest percentile threshold of each of the three competing scores (Tables S4 and S5) averaged 7.0% (SD 3.74%)

- Tables S12 and S13 should have both UKB and MESA columns as well as them combined

We have re-calculated the reclassification statistics in Table S13 and S14 (the numbers have changed slightly with the addition of new Table S5) upon the UK Biobank and MESA datasets separately. Those statistics have been added to the tables in the updated manuscript.

- The authors would be encouraged to make the CAD PRS they develop freely available (i.e. variants and weights) as per the PRS Reporting Standards

As with our previously published CAD PRSs (Bolli et al 2021), we will make the PRSs available for non-commercial use upon request. The manuscript has been modified to communicate this fact.

Line 601

The polygenic risk scores we developed are available for non-commercial use upon request.

REVIEWERS' COMMENTS

Reviewer #2 (Remarks to the Author):

Overall, the concerns raised in the previous round and the study's limitations appear to have been effectively addressed. Additionally, the software's description has become more lucid. While I'm not entirely persuaded by the rationale behind choosing PRS-CSx, the various efforts by the authors, especially their independent work on MetaPRS, lend credence to their choice.

I would like the authors to elucidate the differences and novelty of their manuscript in comparison to the study titled 'A multi-ancestry polygenic risk score improves risk prediction for coronary artery disease' by Drs. Pradeep Natarajan & Amit V. Khera.

Reviewer #2 (Remarks to the Author):

Overall, the concerns raised in the previous round and the study's limitations appear to have been effectively addressed. Additionally, the software's description has become more lucid. While I'm not entirely persuaded by the rationale behind choosing PRS-CSx, the various efforts by the authors, especially their independent work on MetaPRS, lend credence to their choice.

I would like the authors to elucidate the differences and novelty of their manuscript in comparison to the study titled 'A multi-ancestry polygenic risk score improves risk prediction for coronary artery disease' by Drs. Pradeep Natarajan & Amit V. Khera.

We thank the Reviewer for their advice and appreciation of the efforts that we have made to overcome their original concerns.

The recent paper by Patel et al (2022) in Nature Medicine that they mention does indeed cover similar ground to ours. Encouragingly, despite using a different PRS development approach (LDPred), and somewhat different training and validation populations, they also described clear clinical utility of CAD PRS in all ancestries analysed. The main differences between our approach and theirs is that they (a) ultimately generate one PRS panel using a relatively old algorithm that is then applied to any individual, which contrasts with our ancestry-specific panels which are generated from a more modern approach that incorporates multiple different GWASs simultaneously, and (b) integrate the PRS within clinical risk models, whereas we advocate for the use of CAD PRS as a risk enhancer. We believe that both approaches have merit, but have opted to describe the use of PRS as a risk enhancer because this more closely aligns with how additional risk factors - and PRS in particular - are suggested to be used in the proposed changes to AHA guidelines (O'Sullivan et al 2022 Circulation). We nevertheless show in our manuscript the effect of integrating the PRS within the PCE (using our INTEGRATEpce tool) and show roughly equivalent performance to that of Patel and colleagues. So in summary, both papers argue for the use of CAD PRS within clinical risk assessments in multiple ancestries but arrive at these conclusions through different methodological approaches and integration frameworks.